# Spatial Variability of the Mechanical Parameters of High-Water-Content Soil Based on a Dual-Bridge CPT Test

**Haifeng Lu, Huiying Li * and Xiangshuai Meng**

School of Earth and Environment, Anhui University of Science & Technology, Huainan 232001, China; luhaifeng7571@126.com (H.L.); 19966590832@189.cn (X.M.)

* Correspondence: hflu@aust.edu.cn

**Abstract:** Soft soil generally has a high water content, and the accurate quantification of its mechanical parameters is an important aspect of foundation design and disaster prevention. The mechanical parameters of soft soil have significant spatial variability or heterogeneity due to the complex deposition process of soil, leading to the high uncertainty of the quantifications of its parameters. Therefore, understanding the spatial variability of the parameters is an important approach to reduce uncertainty. In this study, the high-resolution (0.1 m) tip resistance ($q_c$) and side friction ($f_s$) of 18 soft soils in coastal areas were measured using the Dual-bridge CPT in-situ test. The vertical and horizontal variabilities of $q_c$ and $f_s$ were investigated using the random field theory. The results showed that both $q_c$ and $f_s$ are stationary and ergodic. The coefficient of variation of vertical $f_s$ is much higher than that of $q_c$. On the one hand, $f_s$ may be vulnerable to noise, and its test accuracy is lower than $q_c$; on the other hand, it may be that the spatial variability of the residual strength of soft soil may be greater than that of its failure strength. The horizontal correlation distance and coefficient of variation of $q_c$ and $f_s$ have no obvious change trend along the depth direction, but compared with the coefficient of variation curve, it was found that the change trends of $q_c$ and $f_s$ are basically the same, which is considered to be related to the properties of the soil layer. The research results can provide support for the spatial variability evaluation and reliability analysis of soft-soil engineering in this area. At the same time, it can also provide a theoretical basis for the layout of exploration engineering and sampling spacing.

**Keywords:** dual-bridge CPT; random field; marine clay; spatial variability; regional characteristics

## 1. Introduction

The soft-soil layer is widely and thickly distributed in China; it has the characteristics of a high water content, low density, low strength, high compressibility, low permeability and medium sensitivity. Among them, marine soft soil has poor engineering properties, such as a high rheological property, a low bearing capacity, poor permeability and the poor uniformity of soil layer. [1–6]. In engineering construction, due to the nature of marine soft soil, engineering design and construction cannot achieve the desired results, resulting in a large number of diseases and security risks, causing engineering accidents. The further strengthening of the research on marine soft soil has important practical significance for the reduction and prevention of engineering accidents. In the study of the soft-soil layer, the accurate acquisition of the soft soil's parameters is a very important prerequisite. As we all know, the physical and mechanical parameters of natural soil have significant spatial variability due to the different depositional environments and later geological function in the formation process, which is manifested in the discreteness and uncertainty of the parameter test values at different spatial locations. Through sampling tests or in-situ tests, only the parameters of soil at limited points in space can be obtained, and its parameter characteristics can't be accurately measured point by point, which

brings inconvenience to the reliability analysis of geotechnical engineering. Vanmarcke (1977) [7] first introduced the random field theory and established the random field model of soil profiling. Through the reduction of point variance, the transition from point characteristics to spatial average characteristics was completed, and the estimation of unknown point parameter characteristics through limited sample points was realized, laying the foundation for the study of the spatial variability of geotechnical parameters by random field theory. With the deepening of the research, the investigation of the spatial variation characteristics of soil parameters has been further developed and improved. At present, a complete system of data processing, stationarity and ergodicity tests, correlation distance calculation and site spatial variability evaluation has been formed [8–13], providing useful spatial variability parameters for geotechnical reliability design.

Soft soil is generally characterized by a high water content, low shear strength and high compressibility, such that it is difficult to obtain an undisturbed sample, which makes it difficult to study the spatial variability of their parameters through laboratory tests. Cone penetration tests (CPT) are a high-precision in-situ testing technique that enable continuous and rapid testing. They are an ideal method to study the parametric random field characteristics of soft soil. For example, Wang et al. (2009), Lin et al. (2015), Guo et al. (2017) and Qu (2021) [14–17] systematically investigated the random-field properties of soft soil along the coast of Guangdong, the sea-phase clay in central Jiangsu, and the coastal clay in Tianjin, China, using the cone resistance of the cone penetration test data, and statistically summarized the spatial variability of the soft soil parameters in the region. Many cone penetration tests on soft ground utilize a double bridge probe that measures the tip resistance $q_c$ and side friction $f_s$. However, the above research results only carried out a random field analysis on cone penetration test $q_c$, and lacked the study of random field characteristics based on side friction $f_s$. The $f_s$ describes the shear strength characteristics of the soil after damage. The ratio of $f_s$ to $q_c$ is often used for the estimation of soft soil sensitivity, and $f_s$ is also often used to estimate the pile side friction. As such, the study of the random field model parameters of $f_s$ is also of practical importance. To this end, this paper takes the coastal soft soil in northern Jiangsu, China, as an example; based on the $q_c$ and $f_s$ data of the cone penetration test, this paper studies its spatial variability using the random field theory, and discusses the random field model parameters and their variation laws in the vertical and horizontal directions, respectively. The research results can provide parameter support for the reliability analysis of soft-soil engineering in this area.

## 2. Random Field Theory

### 2.1. Statistical Characteristics

In the Vanmarcke random field model, the soil profile is regarded as a random function of spatial position coordinates with autocorrelation characteristics, and the spatial variability of the parameters is regarded as a wave component varying around the mean value. When test errors are not considered, the measured soil profile test curve can be characterized by a trend function $t(h)$ and a fluctuation function $w(h)$ together [18], i.e., for a set of soil cone penetration test data $x(h)$, there exists:

$$x(h) = t(h) + w(h) \qquad (1)$$

In the formula, $h$ denotes the sampling depth; the Vanmarcke random field model is based on the weak stationarity assumption, which considers the mean and variance of geotechnical parameters in space as constants independent of spatial coordinates, and its autocovariance function depends only on the distance between two observations [19]. However, in practice, as shown in Figure 1a, due to the influence of geological factors, $x(h)$ tends to show a certain trend with increasing depth, and does not satisfy the basic assumption of being weakly stationary. After a detrending treatment of the geotechnical

parameters, their residuals $w(h)$ can be considered to satisfy the assumption of a zero mean in being weakly stationary [20,21], as shown in Figure 1b.

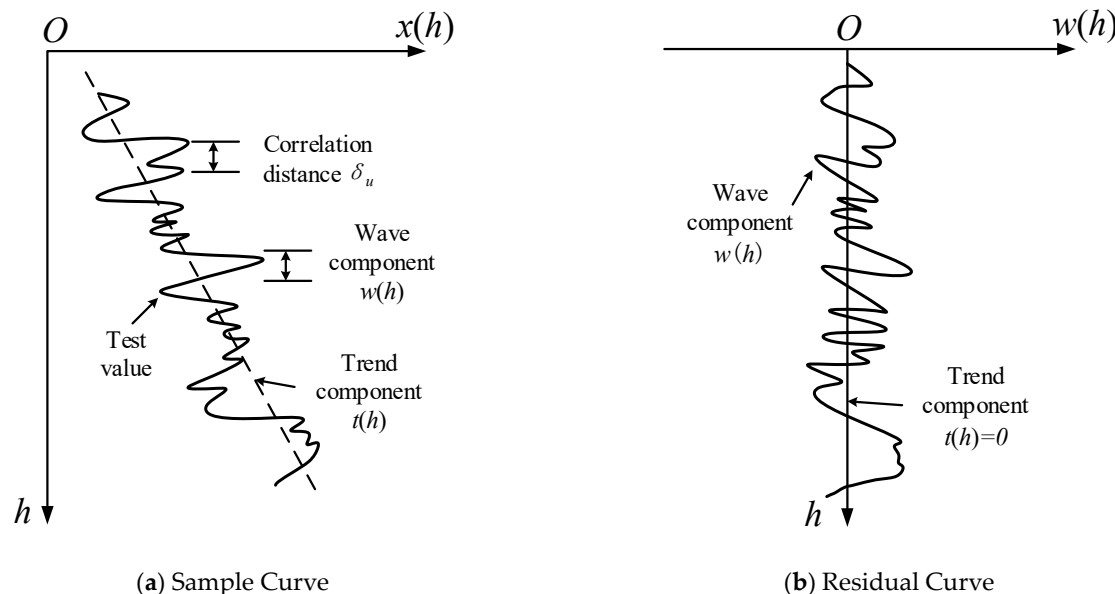

(**a**) Sample Curve                    (**b**) Residual Curve

**Figure 1.** Statistical characteristics of the Wave component and Trend component.

*2.2. Stationarity and Ergodicity of the States*

The knowledge of whether the soil profile conforms to stationarity and ergodicity is a prerequisite for analysis by the random field model. The Vanmarcke random field model simulates the soil profile using a homogeneous normal random field (Gaussian stationary homogeneous random process). Therefore, the data analyzed by the Vanmarcke random field model must conform to the condition of a stationary random field in a mathematical sense. Secondly, the soil mass is a collection of infinite points, and the experimental data are the measured values of individual points. If the analysis results of the experimental points in a borehole are used to reflect the properties of the surrounding soil mass, the data used should have the ergodic properties of various states. It can be seen that whether the spatial distribution of the soil properties is stationary and ergodic is the key to the application of random field method in geotechnical engineering. For soil bodies, the random field model of the soil profile is constructed by using the random field theory, which requires that the random field model must have stationarity and ergodicity [10].

As shown in Figure 2, in the one-dimensional random field $X(h)=\{x_1(h), x_2(h), \cdots\}$, $X_n(h)$ represents the nth sample function of the random field, $n = 1,2...N$. For any depth $h$, the mean value $\mu_X(h)$ of the sample function $X(h)$ is

$$\mu_X(h) = \lim_{N \to \infty} \frac{1}{N} \sum_{n-1}^{N} x_n(h) = E[X(h)] \tag{2}$$

where $N$ is the number of CPT test holes, and $E[X(h)]$ is the mean function of all of the sample functions of the random process $X(h)$ at parameter h, which is called the ensemble average. In the range $(h, h+\tau)$, there exists the autocorrelation function $R_X(h, h+\tau)$:

$$R_X(h, h+\tau) = \lim_{N \to \infty} \frac{1}{N} \sum_{n=1}^{N} x_n(h) x_n(h+\tau) = E[X(h) X(h+\tau)] \tag{3}$$

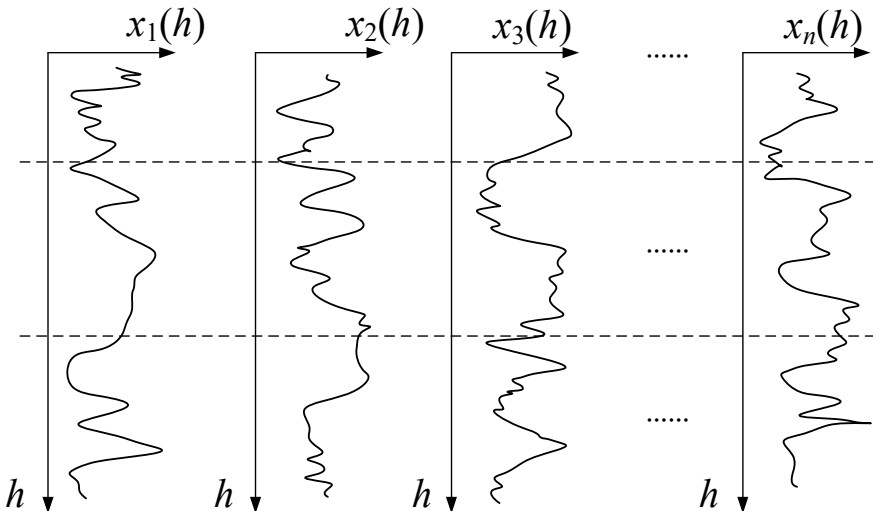

**Figure 2.** One-dimension random field $X(h)$.

If the mean value of the one-dimensional random field sample function is constant, the autocorrelation function is $(h, h+\tau)$, and the random field is said to be stationary. Let $x(h)$ be a mean-square continuous stationary random field, and its average depth $<x(h)>$ will be

$$<x(h)>=\lim_{H\to\infty}\frac{1}{H}\int_0^H x(h)\,dh \tag{4}$$

where, $H$ is the sampling depth of the CPT test hole, if

$$<x(h)>=\mu_X(h)=E\big[x(h)\big] \tag{5}$$

Then the mean value of random field $x(h)$ is ergodicity.

$$<x(h)x(h+\tau)>=\lim_{H\to\infty}\frac{1}{H}\int_0^H x(h)x(h+\tau)\,dh \tag{6}$$

If $<x(h)x(h+\tau)>$ exists, then

$$<x(h)x(h+\tau)>=R_X(h)=E\big[x(h)x(h+\tau)\big] \tag{7}$$

The autocorrelation function of the process is said to have ergodicity [22].

*2.3. Correlation Distance*

In a random field model, if there is

$$\lim_{\tau\to\infty}\sigma\Gamma^{-2}(\tau)=2\lim_{\tau\to\infty}\int_0^\tau\left(1-\frac{\Delta x}{\tau}\right)\rho(\Delta x)\,d(\Delta x)=\delta_u \tag{8}$$

where $\tau=i\Delta x$, $i$ is the multiple of the sampling spacing, $\Delta x$ is the initial sampling spacing, an $\rho(\Delta x)$ is the correlation function. when $\tau$ is large enough, $\sigma\Gamma^{-2}(\tau)$ approaches to a fixed value $\delta_u$; $\delta_u$ is called the correlation distance, and its value represents the autocorrelation distance of the soil parameters. Within the range of the correlation distance, the parameters have correlation, and if the distance is greater than this distance, there is no correlation. At present, there are many methods to solve the correlation distance [23]: (I) the Space Average Method, SAM; (II) the Average zero-span method, VXP; (III) the Bartlett method of sample autocorrelation function, BLM; (IV) and Autocorrelation

function fitting, AMF, etc. The Space Average Method (SAM) and Autocorrelation function fitting (AMF) are the most commonly used methods [20,24].

SAM works through the variance reduction function $\Gamma^2(\tau)$ to solve the correlation distance $\delta_u$. According to the definition of correlation distance, the standard deviation of the whole sample is calculated $\sigma$, a new set of data samples is formed by the mean of adjacent $i$ sample points, and the variance $Var(i)$ of the data is calculated. In this case, the variance reduction coefficient $\Gamma^2(i)$ is:

$$\Gamma^2(i) = \frac{Var(i)}{\sigma^2} \tag{9}$$

Take different $i$ values, repeat the above process, and make the graph of the function $\Gamma(i)-i$, when $i$ takes a certain value, $\Gamma(i)$, which tends to be stable. Find the $\Gamma(i)$ stationary point $n$, and using $\delta_u = n\Delta x\Gamma^2(n)$ find the correlation distance.

AMF uses the autocorrelation function model to fit the autocorrelation coefficient of the fluctuation component of the soil parameter samples and find the correlation distance. In actual calculations, the correlation function $\rho(\tau)$ corresponding to spatial distance $\tau$ is calculated according to Equation (10), and then the fitting method is used to calculate the correlation distance:

$$\rho(\tau) = \rho(i\Delta x_0) = E\left[x(h)x(h+\tau)\right] = \frac{1}{n-i}\sum_{k=1}^{n-i}X(h_k)X(h_{k+i}). \tag{10}$$

Common sample autocorrelation function models include the exponential function (SNX), the linear exponential correlation function (LNX), the exponential cosine function (CSX), and the linear-exponential-cosine model (LNCS), as shown in Table 1 [5,19,25,26].

**Table 1.** Correlation function and correlation distance.

| Correlation Function | Mathematical Expression | Correlation Distance $\delta_u$ |
|---|---|---|
| SNX | $e^{-a|\tau|}$ | 2/a |
| LNX | $(1 + a|\tau|)e^{-a|\tau|}$ | 4/a |
| CSX | $e^{-a|\tau|}\cos(a|\tau|)$ | 1/a |
| LNCS | $(1 + a|\tau|)e^{-a|\tau|}\cos(a|\tau|)$ | 1/a |

It is worth pointing out that in geotechnical investigation, the sampling spacing in the horizontal direction is usually large, and it is difficult to obtain a large volume of sample data, such that the horizontal random field model and parameters cannot be calculated by the strict autocorrelation function fitting method. Therefore, the method to determine the horizontal random field parameters of soil is different from that of the vertical random field parameters of soil. VXP uses the average length $\overline{d}$ of the intersection of the soil parameters function curve $r(h)$ and its trend function curve $t(h)$ to solve the horizontal fluctuation range $\delta_h$. It is commonly used to solve the correlation distance under limited boreholes. The calculation formula is:

$$\overline{d} = \sum_{i=1}^{n}d_i/n \tag{11}$$

$$\delta_h = \sqrt{\frac{2}{\pi}}\overline{d} \approx 0.8\overline{d} \tag{12}$$

where $d_i$ is the segment length, as shown in Figure 3.

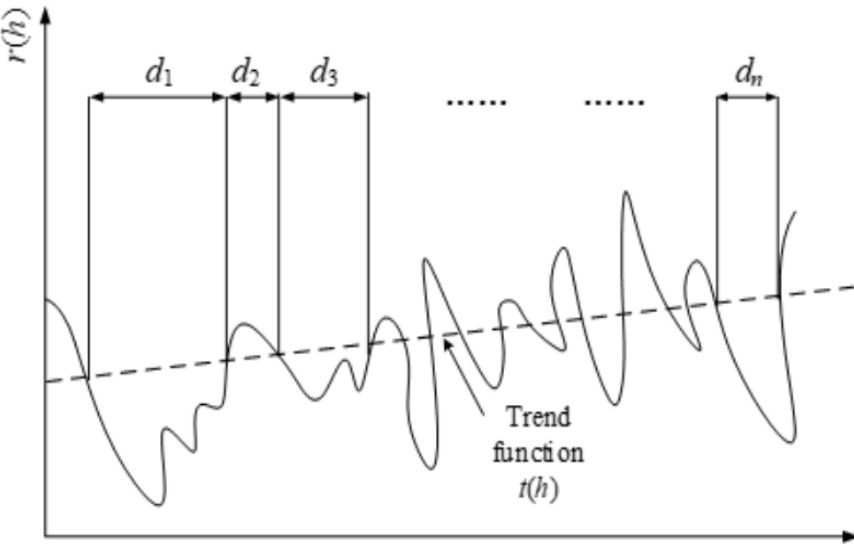

**Figure 3.** Average zero-span method VXP.

*2.4. Coefficient of Variation*

The coefficient of variation is one of the important indicators to characterize the internal variability of soil layers. Phoon and Kulhawy suggested the use of the inherent coefficient of variation $Cov_w$ instead of sample variance to characterize the internal variability of soil [9], and the calculation formula is

$$Cov_w(h) = \frac{\sigma_w(h)}{t(h)} \tag{13}$$

where $t(h)$ is the trend item of the soil parameter test data, and $\sigma_w(h)$ is the standard deviation of the fluctuation component of the same soil parameter.

**3. Application and Discussion**

*3.1. Project Profile*

The Guanhe River embankment project in Guannan County, Lianyungang City, Jiangsu Province is the research object, as shown in the Figure 4; the site is located in the north of the Northern Jiangsu Plain, close to the Yellow Sea. The ground elevation is 2–3 m, the terrain is relatively flat, and the landform characteristics belong to marine plain accumulation landforms. The main structural feature in the site is the Cathaysian Huaiyin–Xiangshuikou fault, which extends into the Yellow Sea at an angle of 35–45°. No signs of activity have been found in the late and recent times, and the regional geological stability is good. The site is based on the epimetamorphic rock series in the mesoproterozoic area, the main cover layer is composed of the Sinian system to the Triassic system, and the upper quaternary system is mainly a marine-deposit soil layer. The silt layer of the marine deposit in the east of the site gradually thickens.

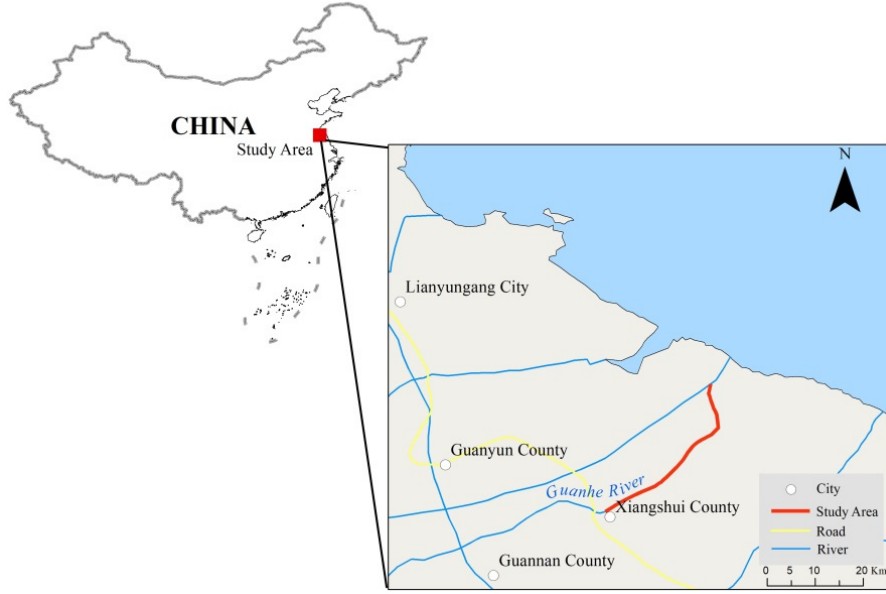

**Figure 4.** Study area location.

The boreholes and CPT holes for the geotechnical investigation and construction are mainly arranged along the Guanhe River embankment, with a total of 12 coring boreholes and 18 CPT holes. The physical and mechanical properties of the strata and each layer divided according to the site CPT test data and drill core data are shown in Table 2. As can be seen from the table, the soil layer of site $2_1$ is mucky soil, which is close to the surface and mainly affected by marine accumulation. The thickness of the layer is great, with an average thickness of 16.3 m. It has the characteristics of high compressibility, low strength, high water content and high sensitivity; it is the main soil layer affecting the stability of the embankment. At the same time, the soft-soil layer is also the main object of this paper.

**Table 2.** Main soil characteristics of the field.

| Stratum Number | Geotechnical Name | State | Thickness (m) | Buried Depth on Roof m | Cohesion (Consolidated Quick Shear) (KPa) | Internal Friction Angle (Consolidated Quick Shear) (°) | Compression Modulus (MPa) | Blow Count of spt N63.5 |
|---|---|---|---|---|---|---|---|---|
| A | plain fill | soft plastic-plastic | 0.3–4.9 | 0 | 25.4 | 12.4 | 4.11 | 7 |
| $1_2$ | clay | soft plastic-plastic | 0.2–2.5 | 0.3–4.9 | 24.0 | 12.0 | 3.75 | 5 |
| $2_1$ | mucky soil | flow plastic | 8.9–20.4 | 0.5–10.40 | 17.7 | 9.1 | 2.67 | 1 |
| $2_1$ * | silty loam | Loose-slightly dense | 2.0–6.3 | 1.60–4.80 | 13.6 | 19.4 | 6.79 | 8 |
| 3 | heavy silty loam | Plastic-soft plastic | 2.0–3.1 | 11.90–16.40 | 22.8 | 14.3 | 4.79 | 6 |
| $4_1$ | silty clam | plastic | 0.8–4.8 | 15.00–25.70 | 34.2 | 14.6 | 5.49 | 12 |
| $4_1$ * | silty loam | slightly dense-medium density | – | 16.70–28.50 | 6.4 | 24.9 | 11.97 | 19 |

The CPT test adopts a standard double bridge probe, which can test the tip resistance $q_c$ and side friction $f_s$. The probe penetrates into the soil by the static pressure method, the penetration rate is 1.2 m/min, the penetration reading interval is 0.1 m, the field zeroing error is less than 3%, the average spacing of CPT holes is 18.7 m, the hole layout position is shown in the Figure 5, and the test curve of typical CPT test holes and soil stratification results are shown in the Figure 6.

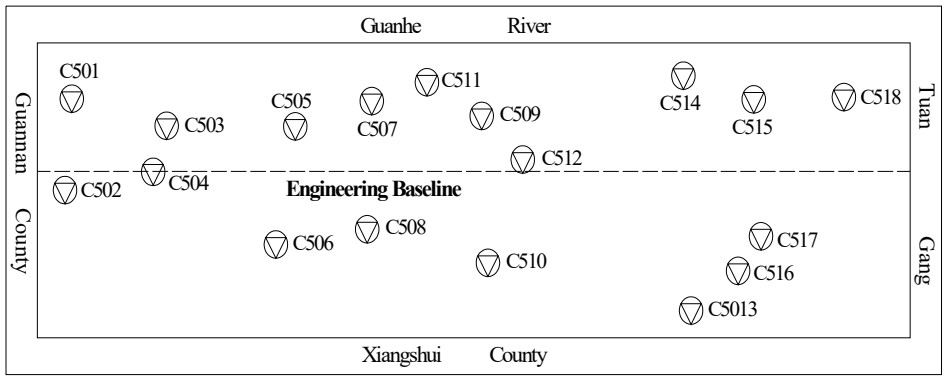

**Figure 5.** Layout plan of a CPT test hole.

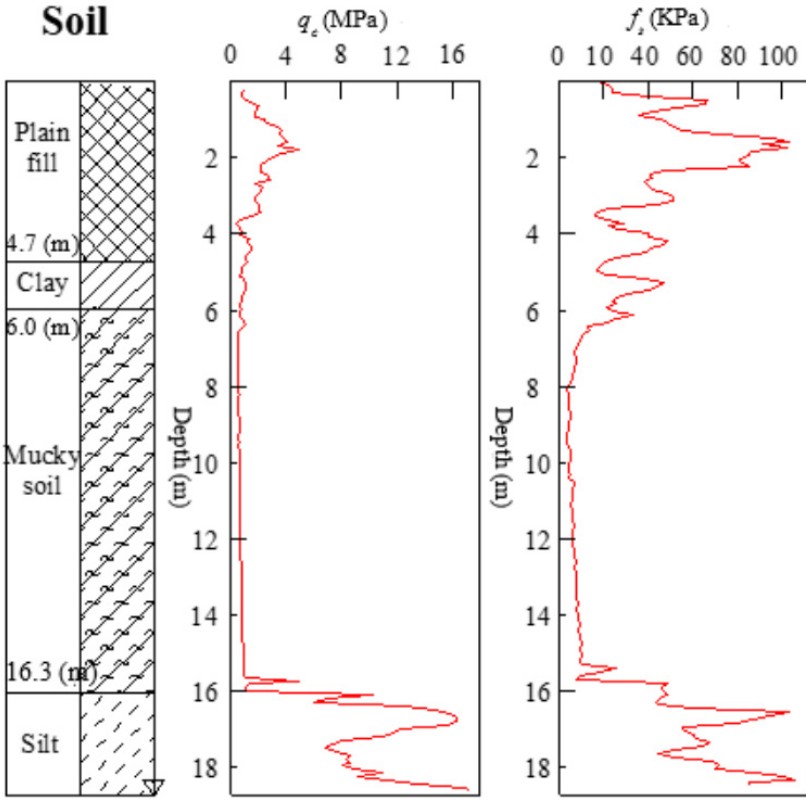

**Figure 6.** Test curve of typical CPT test holes and layers.

### 3.2. Outlier Test

For a group of soil sample test data, due to sampling methods, test methods and other reasons, abnormal data will inevitably appear, which affects the representativeness and authenticity of the data. Therefore, the test data must be checked for outliers. There are many methods in outlier testing: the $3\sigma$ test, Grubb's test, the T-test, and the jump degree test, etc. [24,27–29]; the $3\sigma$ test is one of the most widely used methods. The basic steps are the determination of the mean $\mu$ and standard deviation $\sigma$ of the sample, the construction of a valid data range $[\mu - 3\sigma, \mu + 3\sigma]$, and the testing of the sample values one by one; outside the interval range is abnormal data, which should be discarded.

The tip resistance $q_c$ and side friction $f_s$ of CPT hole C514 in the site were taken as examples and tested by the $3\sigma$ test method. In order to visually illustrate the valid range of the data, we made $3\sigma$ graphs of $q_c$ and $f_s$, as shown in Figure 7. As can be seen from the graph, the $3\sigma$ test shows that the tip resistance $q_c$ data of the C514 exploration hole were within the valid range, and there were no abnormal data, while the abnormal data of $f_s$ were deleted. The same method was used to test the abnormal values of other boreholes, which provided effective data support for subsequent random field analysis.

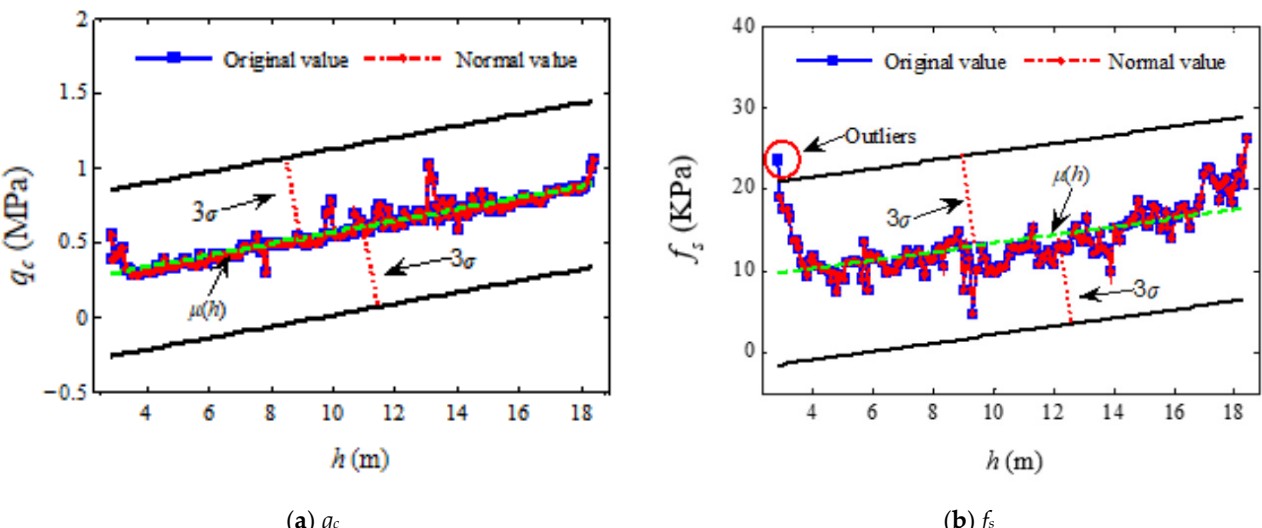

(**a**) $q_c$          (**b**) $f_s$

**Figure 7.** C514 exploratory hole, $3\sigma$ method outlier test.

### 3.3. Trend Removal Processing

The tip resistance $q_c$ and side friction $f_s$ obtained by the CPT had a clear trend along the depth. In order to satisfy Vanmarcke's assumption of the weak stability of a random field, it was necessary to detrend the original data [30]. Numerous studies have found that the trend component functions of the soil parameters are mostly linear, with a few higher-order functions, and suggested that the highest choice of trend function is no more than a quadratic nonlinear function [19,20]. Similarly, taking the C514 CPT test hole data as an example, linear and quadratic polynomials were used to fit the trend function, respectively, as shown in Figure 8. It can be seen that the fitting results of the two methods of $q_c$ are basically the same. In order to facilitate a calculation, the linear trend function was selected to determine that the trend term of the tip resistance was $q_c = 0.0383\,h + 0.1888$. For $f_s$, it was found that the quadratic nonlinear function fits better than the linear function, and its trend item was $f_s = 0.100\,h^2 - 1.56\,h + 16.94$. We can then subtract the trend item from the original value to get the data curve of the detrended fluctuation item. Figure 9 shows the detrended result of this hole.

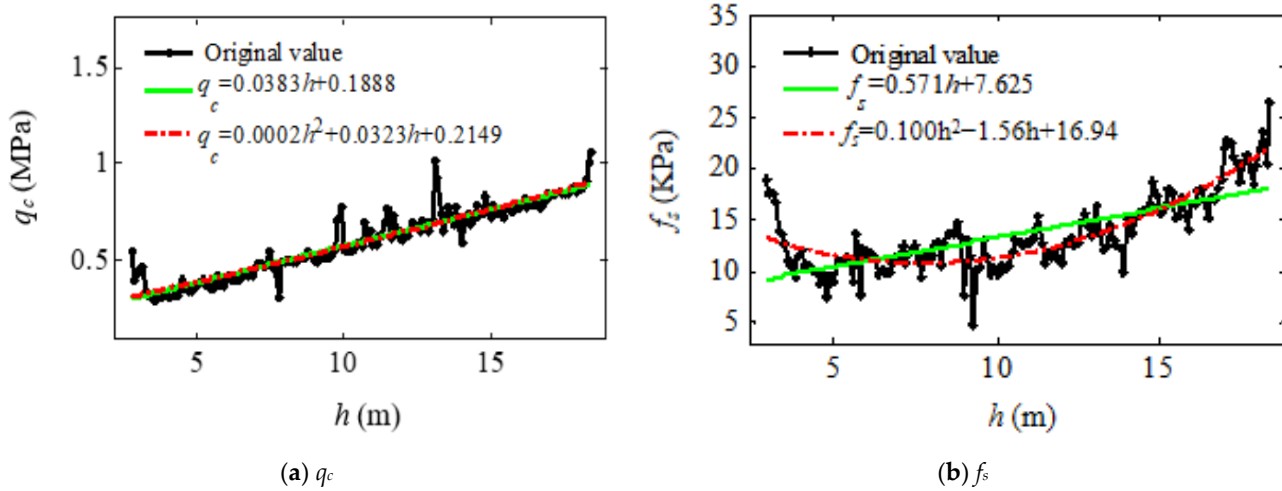

**Figure 8.** Primary fitting and quadratic fitting.

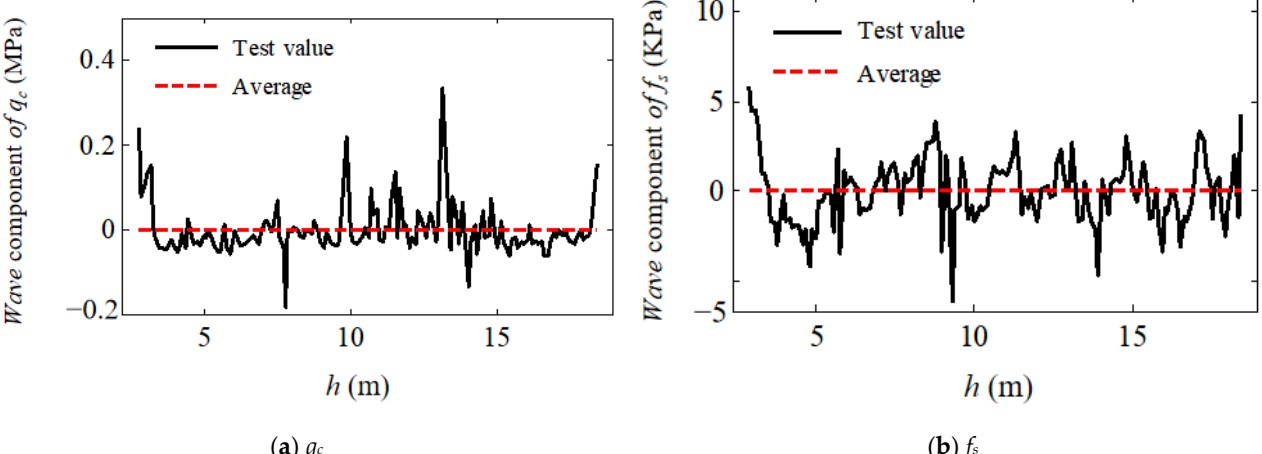

**Figure 9.** C514 trend removal results.

It can be seen from Figure 9 that the data of the tip resistance and side friction after trend removal fluctuated up and down on the axis of 0, only showing the volatility of the data, and the trend removal effect was good. The $q_c$ and $f_s$ data of other CPT holes underwent trend removal by the same method. Based on this, the fluctuation components (residuals) of $q_c$ and $f_s$ of all of the CPT test holes of marine mucky soft soil in the site could be obtained.

*3.4. Stationarity and Ergodicity Tests*

Based on the $q_c$ and $f_s$ data of 18 CPT holes of the marine soft-soil layer, i.e., soil layer $2_1$, and the test principle of random field stationarity and ergodicity in the previous section, the relevant programs were compiled by Matlab 7.0 software. The results were as follows:

From Figures 10 and 11, it can be seen that the ensemble average and correlation functions of the tip resistance $q_c$ and side friction $f_s$ of the marine mucky soft-soil layer at the site did not vary significantly along the depth direction, indicating that each sample of the soft soil random field did not change with the depth in the sense of probability, and the random field of soil prodile has the stability. $q_c$, $f_s$ and the depth correlation function

have no change along the horizontal direction, indicating that the mean values of each sample function and the depth correlation function of the tested random field do not change with the horizontal distance in a significance of the probability, that is, the random field of the two parameters of the soil profile have ergodicity of each state, which can meet the requirements of random field modeling.

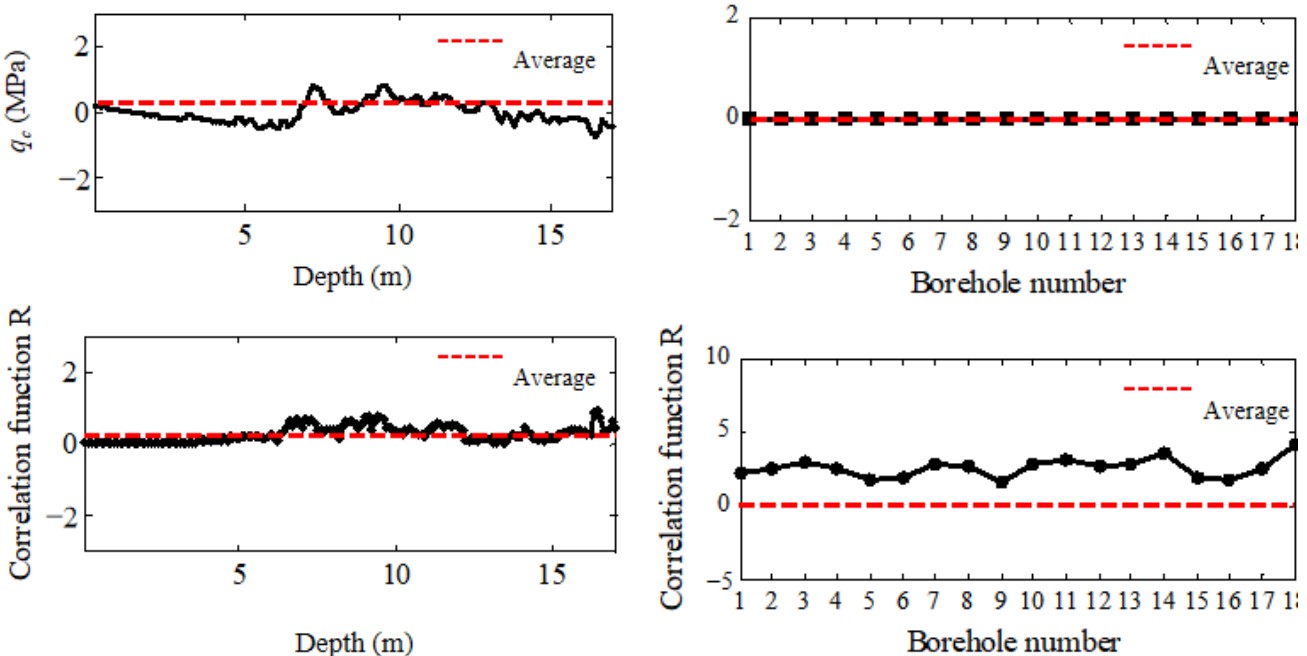

**Figure 10.** Tip resistance $q_c$ stationarity and ergodicity tests.

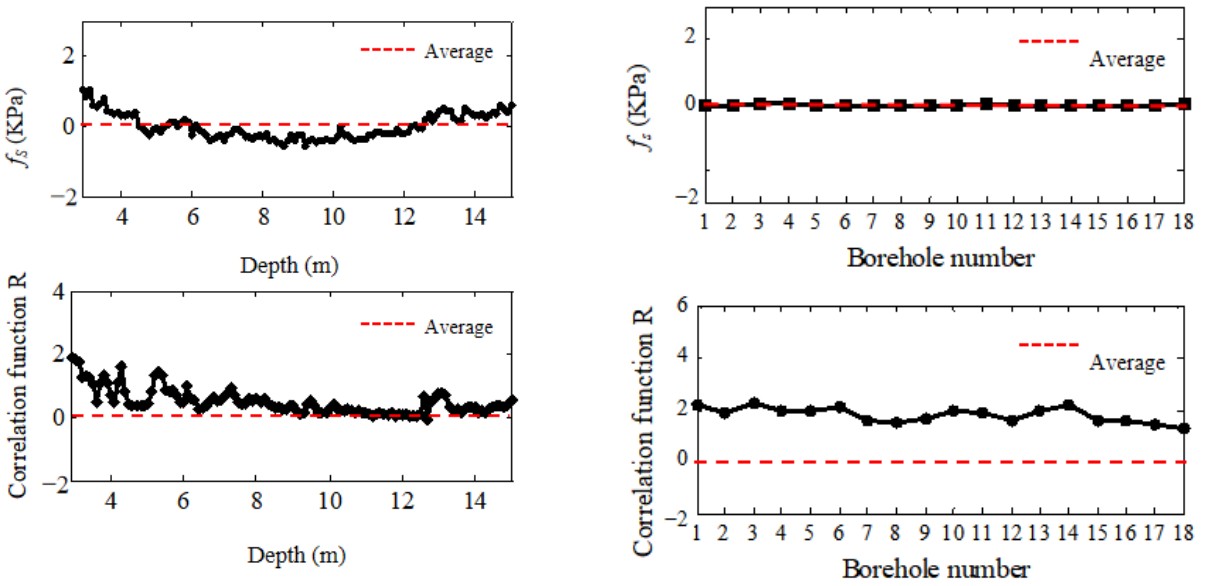

**Figure 11.** Side friction $f_s$ stationarity and ergodicity tests.

### 3.5. Vertical Correlation Distance Solution

The stability and ergodicity of the random field of the soil profile show that the muddy soft-soil layer can be analyzed using the random field model. Based on the random field theory introduced above, the commonly used SAM and AMF were used, respectively. Taking the $q_c$ and $f_s$ data of the C514 CPT drills' silty soft soil as examples, the calculation of the vertical correlation distance was introduced in detail. For the solution of the recursive space method, the sampling interval was set as $i\Delta z$. $\Delta z$ is the initial sampling interval of 0.1 m, and $I = 1–10$. The calculation results are shown in Figure 12.

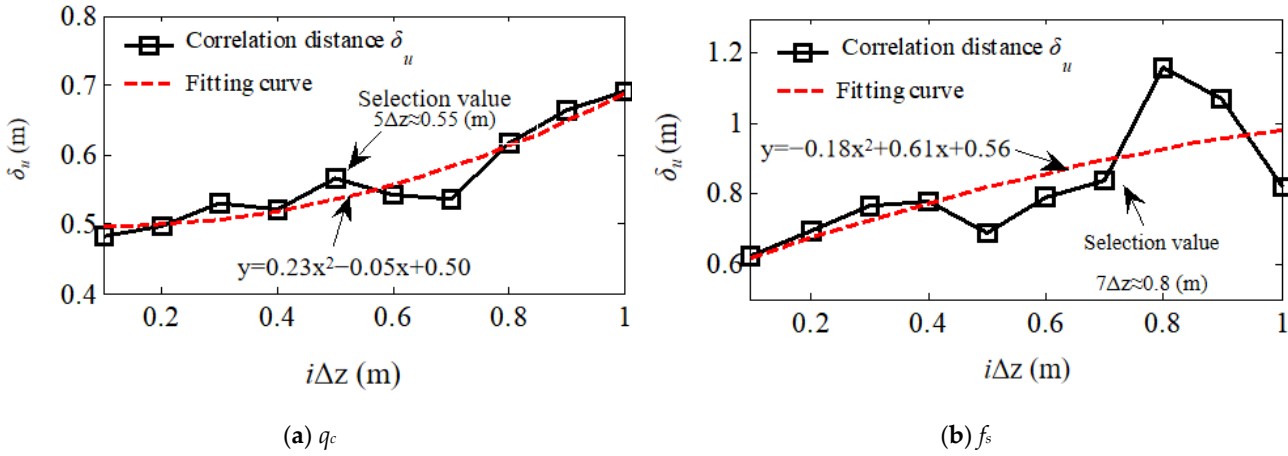

**(a)** $q_c$                                                                  **(b)** $f_s$

**Figure 12.** Vertical correlation distance of the parameters of the muddy soft-soil layer.

It can be seen from Figure 12 that when the sampling spacing is small, the vertical correlation distance calculated by different sampling spacing changes little; with the increase of the sampling spacing, due to the decrease of data points, the volatility of the calculation results based on $q_c$ and $f_s$ data increases, such that the variability of the soil parameters is homogenized, and the reliability of the correlation distance is greatly reduced, which cannot characterize the autocorrelation characteristics of real soil. According to the previous research results, when $i\Delta z \approx \delta_u$, the calculated correlation distance is more appropriate as the soil autocorrelation distance [31]. Thus, for $q_c$, select $5\Delta z$ as the correlation distance corresponding to it is 0.55 m; for $f_s$, select $7\Delta z$, as the correlation distance corresponding to it is 0.8 m.

In order to verify the calculation results of the recursive space method, the four autocorrelation function models described in the previous section (Table 2) were used to fit the $q_c$ and $f_s$ correlation functions. The function fitting results are shown in Figure 13.

From Figure 13, it can be seen that the two parameter correlation functions of $q_c$ at 3 m and $f_s$ before 5 m fit well, and after the above values there is a greater volatility and a poorer fitting effect. Some scholars suggested taking the first half of the correlation function of the soil layer for fitting after research [15,22,28]. Therefore, in this paper, the soil layer in the depth directions of 0–2.2 m and 0–5 m were used to fit the four functions of $q_c$ and $f_s$, respectively. The results show that the two parameters were SNX functions with the best fitting effect and LNCS function with the worst fitting effect. It was concluded that the SNX fitting correlation function expression of $q_c$ is $y = e^{-3.352|\tau|}$, and the vertical correlation distance $\delta = 2/3.352 = 0.597$ m. The expression of the SNX fitting correlation function of $f_s$ is $y = e^{-2.423|\tau|}$, and the vertical correlation distance $\delta = 2/2.423 = 0.825$ m. Compared with the SAM, the calculation results are larger, which is in agreement with the research results of [25]. This indicates that the method of calculating the vertical

correlation distance by $q_c$ and $f_s$ in this paper is reliable, and can provide a theoretical basis for the further analysis of the fluctuation range of the soft-soil layer in the site.

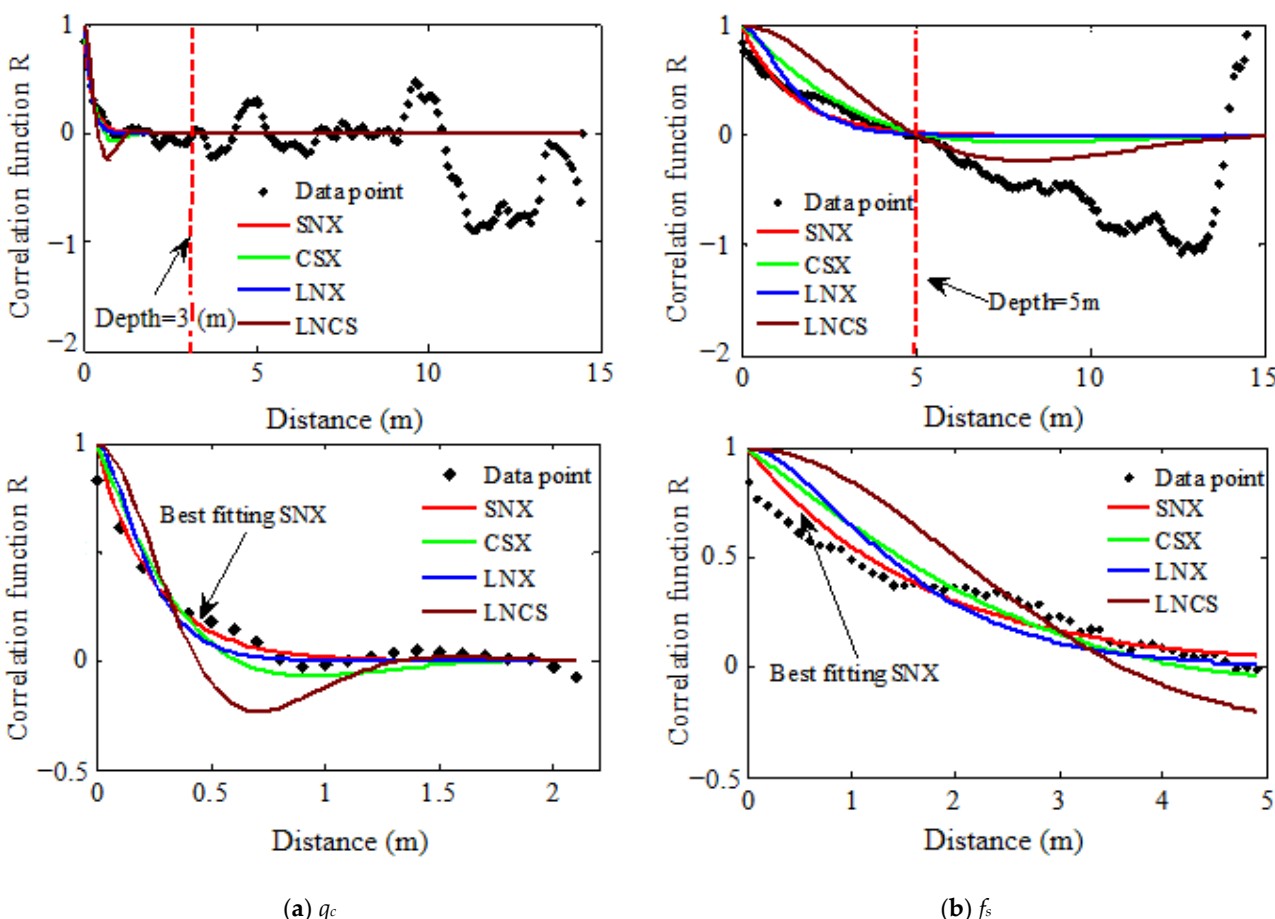

**(a)** $q_c$                                                  **(b)** $f_s$

**Figure 13.** Correlation function fitting of silty soft-soil parameters.

In order to reasonably determine the vertical correlation distance of the marine mucky soft-soil layer, and to obtain the calculation results with statistical characteristics, the SAM and the correlation distance method were used to calculate the results, respectively, according to the $q_c$ and $f_s$ data of 18 CPT holes in the site. The results are shown in Table 3. It can be seen from Table 3 that under the condition of multiple samples, no matter $q_c$ or $f_s$, the calculation result of the vertical correlation distance of the correlation function method is still larger than that of the SAM, but the standard deviation of the SAM is smaller than that of the correlation function method. The fluctuation range of the $q_c$ vertical correlation distance of marine mucky soft soil is 0.100–0.624 m, the average value is 0.324 m, and the coefficient of variation is 38.9%. The fluctuation range of the $f_s$ vertical correlation distance is 0.188–0.942 m, the average value is 0.386 m, and the coefficient of variation is 62.8%.

**Table 3.** Statistics of the vertical correlation distance of the soil layers.

| Soil Parameters | Average | Computing Method | δ fluctuation Range | δ Average (me) | δ Standard Deviation | Coefficient of Variation |
|---|---|---|---|---|---|---|
| $q_c$ | 0.374–0.842 (MPa) | SAM | 0.100–0.624 | 0.324 | 0.833 | 38.9% |
| | | Correlation function method | 0.153–0.785 | 0.573 | 1.785 | 32.1% |
| $f_s$ | 9.13–23.24 (KPa) | SAM | 0.188–0.942 | 0.386 | 0.620 | 62.8% |
| | | Correlation function method | 0.163–0.836 | 0.433 | 0.651 | 66.5% |

For the marine soft soil, in this case, the coefficient of variation of $f_s$ is much higher than that of $q_c$. On the one hand, it may be that $f_s$ is vulnerable to noise and its test accuracy is lower than that of $q_c$, such that it shows high variability [26,32]; on the other hand, it may be that the spatial variability of the residual strength of soft soil may be greater than that of the failure strength. Of course, this speculation needs further verification. In addition, the variation ranges of both the $q_c$ and $f_s$ vertical correlation distances are larger than the fluctuation ranges of the coastal marine clays in central Jiangsu proposed by [14], reflecting that the spatial variabilities of the marine soft clays in central and northern Jiangsu differ greatly, and the geographical characteristics of both are obvious. At the same time, the coefficient of variation is also much higher than that of the central marine clays, showing higher variability characteristics. Comparing the physical and mechanical properties and the depositional environment of the soft soil in both, the depositional environment of the soft soil in the central and northern coastal areas of Jiangsu is different. From the Lianyungang Area in the north to the Yancheng area in the middle, and then to the Nantong area in the south, the sedimentary facies are littoral facies, lagoon facies and neritic facies in turn. The buried depth of the soft soil decreases from shallow to deep, the porosity ratio and water content decrease in turn, and the density increases in turn, showing obvious different characteristics.

Based on the vertical correlation distance of the soil layer above, the sampling interval of the rock coring test can be guided; that is, the sampling interval should be equal to or slightly less than the vertical correlation distance of the corresponding rock formation.

*3.6. Horizontal Correlation Distance Solution*

The horizontal correlation distance is also an important indicator to describe the spatial variability of soil. However, due to the fact that most CPT holes are not on a straight line and the spacing between the two holes is unequal, it is difficult to obtain enough and effective statistical samples, and the level random field model and parameters cannot be calculated by the strict autocorrelation function fitting method [33–35]. In view of this, VXP is used to calculate the horizontal correlation distance. VXP is suitable for the estimation of geotechnical parameters with a small parameter sample size, and its calculation results are similar to those of other methods. According to the previous work, it is considered that the test data of CPT holes with a radial distance of less than 20 m between the two holes are valid [35], and 7 of the 18 static exploration holes were selected as the exploration holes for the calculation of the horizontal correlation distance.

The data of $q_c$ and $f_s$ from seven boreholes at depths of −6.50 m and −10.5 m in the study soil layer were taken, respectively, and the correlation distances were calculated using the VXP, as shown in Figure 14.

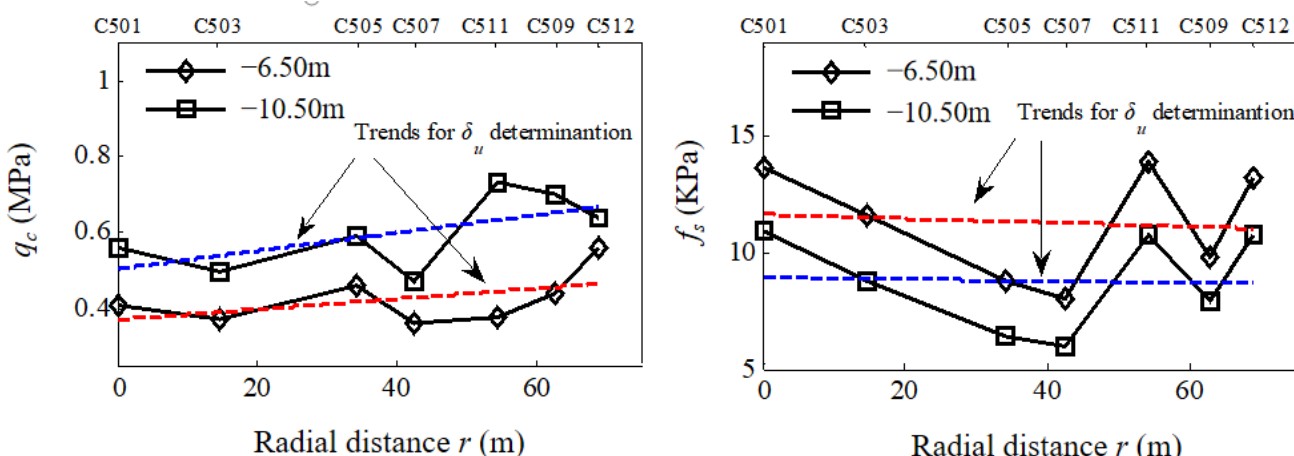

**Figure 14.** Radial distribution of the $q_c$ and $f_s$ representative samples for the determination of the horizontal correlation distance.

As shown in Figure 14, in this case, the horizontal correlation distance of $q_c$ at the depth of −6.50 m was (63.0 − 9.6) × 0.8/3 = 14.2 m, and the $f_s$ horizontal correlation distance was (66.3 − 15.1) × 0.8/3 = 13.7 m. The horizontal correlation distance of $q_c$ at the depth of −10.50 m was (66.7 − 8.8) × 0.8/3 = 15.4 m, and the $f_s$ horizontal correlation distance was (63.8 − 14.2) × 0.8/3 = 13.2 m. The number of intersections between the sample curve and the trend function was greater than two, and the calculation results were reliable.

Lin et al. (2015) studied the variation trend of the horizontal correlation distance of Jiangsu marine clays with depth, and its relationship with the horizontal variation coefficient in the literature. It was concluded that the variation range of the horizontal coefficient of variation of Jiangsu marine clays was 5.0–11.1 m, with an average of 7.8 m, and the variation range of the horizontal coefficient of variation was 11.5–39.0%, with an average of 19.86%. The same method was used to analyze the horizontal spatial variability of the tip resistance $q_c$ and side friction $f_s$ of mucky soil $2_1$ in this example. All of the test data of seven boreholes in the study soil layer were taken at an interval of 0.1 m. The calculation results are shown in Figure 15.

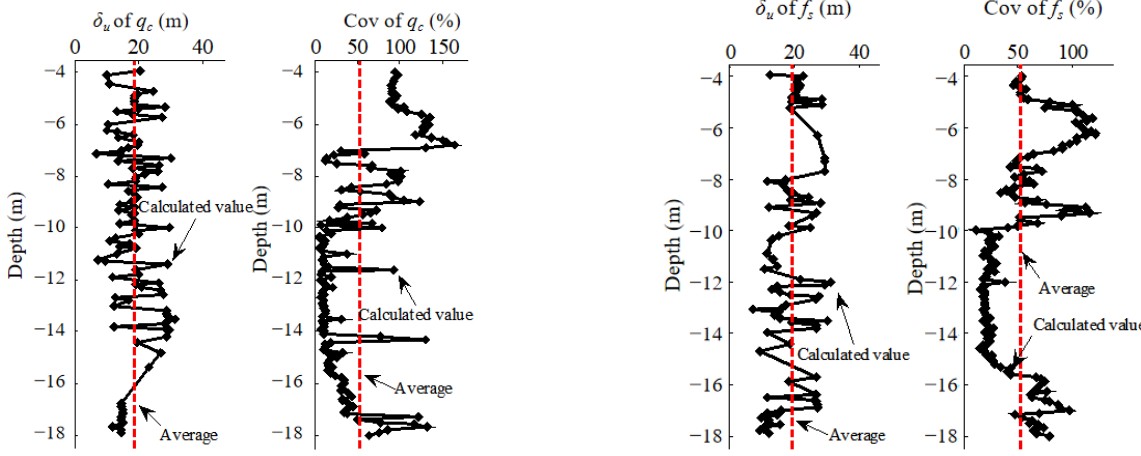

**Figure 15.** Horizontal spatial variability of tip resistance $q_c$ and side friction $f_s$.

According to the calculation, the average horizontal correlation distance of mucky soil tip resistance is 19.18 m, the fluctuation range is 6.02–35.05 m, the coefficient of variation range is 5.8–163.5%, the average value is 53.8%, the average horizontal correlation distance of mucky soil side friction is 20.32 m, the fluctuation range is 7.23–32.27 m, the coefficient of variation range is 11.3–121.5%, and the average value is 52.6%; the horizontal correlation distance and coefficient of variation of the two parameters are basically the same, and have strong consistency. As can be seen from Figure 15, the variation range of the horizontal fluctuation range of $q_c$ and $f_s$ is large on the whole, both of which are greater than the reported values, showing strong variability and different regional characteristics. Moreover, the horizontal correlation distance and coefficient of variation of $q_c$ and $f_s$ have no obvious change trend along the depth direction. There is no significant correlation between the horizontal correlation distance of the two parameters and the coefficient of variation curve. However, by comparing the coefficient of variation curve, it was found that the variation trends of $q_c$ and $f_s$ are basically consistent, which is related to the soil properties. The coherence and variability of soil properties determine the consistency and variability of soil parameters. It should be noted that due to the limitation of the sampling spacing, the determination of the horizontal correlation distance of coastal soft soil in Northern Jiangsu is only based on relatively few exploration holes; the calculation result has a certain uncertainty, and can only be used as an approximate result. The results show that the parameters of the soil random field model are closely related to the observation scale, and this size effect should not be ignored in specific engineering applications. When the observation scale is too small, the statistical results have a great fluctuation, which is affected by the test error; when the observation scale is too large, the statistical results are affected by the geological environment, and the characterization of the soil spatial variability becomes weaker.

## 4. Conclusions

This paper systematically studied the evaluation method of the spatial variability of geotechnical parameters based on random field theory. Taking the marine mucky soil layer of the Guanhe embankment project in Guannan County, Lianyungang City, Jiangsu Province as an example, based on the test data from the CPT of a double bridge, the spatial variability was analyzed by random field theory, and the following conclusions were drawn:

1. Taking the marine mucky soil layer 21 of exploration hole C514 as an example, the $3\sigma$ rule was used to test the soil data for outliers, and the test results were good. Comparing the linear and non-linear fitting, the linear function was selected as the trend item for trend removal, and the processed data could be used to construct a random field model for the site soil layer.

2. The stationarity and ergodicity of the tip resistance $q_c$ and side friction $f_s$ data of mucky soil layer 21 were tested. The results show that the two parameters of the site soil layer had stationarity and ergodicity. Based on the SAM, the vertical correlation distances of tip resistance $q_c$ and side friction $f_s$ were 0.324 m and 0.386 m, respectively. The average coefficients of variation were 38.9% and 62.8% respectively. The horizontal correlation distances of tip resistance $q_c$ and side friction $f_s$ obtained by VXP were 19.18 m and 20.32 m, respectively, and the average coefficients of variation were 53.8% and 52.6%, respectively.

3. The variation coefficient of $f_s$ in the vertical direction is much higher than that of $q_c$, and the correlation distance and variation coefficient in the horizontal direction are very consistent. Both of them show strong variability and different regional characteristics.

4. The borehole coring and borehole layout are very important for engineering investigation. The vertical and horizontal correlation distances have guiding significance for the borehole coring interval and borehole layout interval. In the project

site, the sampling interval of the rock coring test will be equal to or slightly less than the vertical correlation distance of the corresponding rock stratum. The spacing of the holes will be equal to or slightly less than the horizontal distance of the corresponding rock stratum.

**Author Contributions:** Conceptualization, Haifeng Lu and X.M.; Funding acquisition, Haifeng Lu; Investigation, Haifeng Lu and X.M.; Methodology, Haifeng Lu; Validation, X.M.; Visualization, X.M. and Huiying Li; Writing—original draft, Haifeng Lu and X.M.; Writing—review and editing, X.M. and Huiying Li All authors have read and agreed to the published version of the manuscript.

**Funding:** This study was partially supported by research grants from the major projects of natural science research in the Higher Education Institutions of Anhui Province (KJ2019ZD11), the Open Fund of State Key Laboratory of Coal Resources and Safe Mining (No. SKLCRSM20KFA06), and the National Natural Science Foundation of China (41977253). The data used in this paper can be accessed by contacting the corresponding author directly.

**Institutional Review Board Statement:** Not applicable.

**Informed Consent Statement:** Not applicable.

**Data Availability Statement:** The data used in this paper can be accessed by contacting the corresponding author directly.

**Acknowledgments:** This study was partially supported by research grants from the major projects of natural science research in the Higher Education Institutions of Anhui Province (KJ2019ZD11), the Open Fund of State Key Laboratory of Coal Resources and Safe Mining (No. SKLCRSM20KFA06), and the National Natural Science Foundation of China (41977253). The data used in this paper can be accessed by contacting the corresponding author directly.

**Conflicts of Interest:** The authors have no conflict of interest to declare.

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
