# Peer review of "Spatial Variability of the Mechanical Parameters of High-Water-Content Soil Based on a Dual-Bridge CPT Test"

_water, doi:10.3390/w14030343_

Round 1
Reviewer 1 Report
This paper from the perspective of soft soil with high moisture content spatial variability, the tip resistance (qc) and side friction (fs) are studied by random field theory and the spatial variation characteristics of marine soft soil are determined. The paper is informative, well-structured, with clear objectives and conclusions of scientific interest, but a few revisions required. Please see the comments below.
General Comments:
- It is suggested to add a location map of the study area in the application and discussion section. It is suggested that the properties of marine soft soils should be analyzed and described as necessary, and the differences between marine soft soils and terrestrial soft soils should be briefly explained.
- The summary and conclusion should be further condensed to highlight the key points.
- The annotation of references in this paper should be standardized, and all references cited should be marked. Some references in the introduction part of the article are not marked.
- The content of the stationarity and ergodic tests of each state is not sufficient. Explain why stability and ergodicity tests should be performed.
- It is suggested to add the correlation analysis on the engineering properties of marine soft soil with high water moisture content and its influence on actual engineering, so as to make the research purposemore scientific and practical.
Minor comments:
Line 12: Please clarify this sentence;
Line 101-103: These notations should be consistent with the formula;
Line 123: It might be that the sentence needs revising because you intended to say something slightly different.
Line 140-141: Please describe this accurately.
Line 220: Here are two pictures.
Author Response
Dear Reviewer, thank you for reviewing our manuscript and for the constructive comments, which greatly helped us to improve the manuscript. The manuscript was carefully revised and point-by-point response was listed below. We hope that your comments have been addressed accurately.
Question 1: It is suggested to add a location map of the study area in the application and discussion section. It is suggested that the properties of marine soft soils should be analyzed and described as necessary, and the differences between marine soft soils and terrestrial soft soils should be briefly explained.
Answer: Thanks for your suggestion, for the location map of the study area, we have made Study area map to add to the paper page 6 line 201-202. About the properties of marine soft soils and the differences between marine soft soils and terrestrial soft soils, after the study of relevant materials, we have modified it in the introduction. The specific content is “Soft soil layer is widely and thickly distributed in China, it has the characteristics of high water content, low density, low strength, high compressibility, low permeability and medium sensitivity. Among them, marine soft soil has poor engineering properties, such as high rheological property, low bearing capacity, poor permeability and poor uniformity of soil layer.” It's on page 1 of the article, lines 28-31.
Question 2: The summary and conclusion should be further condensed to highlight the key points.
Answer: Thank you for your advice. We have further simplified and emphasized the summary and conclusion. Conclusion 3 is simplified and emphasized, and the specific content is modified as “The variation coefficient of fs in the vertical direction is much higher than that of qc, and the correlation distance and variation coefficient in the horizontal direction are very consistent. Both of them show strong variability and different regional characteristics.” It's on page 15 of the article, line 490-493.
Question 3: The annotation of references in this paper should be standardized, and all references cited should be marked. Some references in the introduction part of the article are not marked.
Answer: This advice is of great importance, We have rearranged the references of the article and mark all references, added and marked “Wang et al. (2009), Lin et al. (2015), Guo et al. (2017) and Qu (2021)” is [14-17], in page2 line57.
Question 4: The content of the stationarity and ergodic tests of each state is not sufficient. Explain why stability and ergodicity tests should be performed.
Answer: Thanks for your suggestion, As for the reasons for conducting stability and ergodicity tests, we have consulted relevant materials and made supplementary modifications on page 3 line 98-107 of the article, the specific content is “Vanmarcke random field model simulates soil profile by homogeneous normal random field (Gaussian stationary homogeneous random process). Therefore, the data analyzed by Vanmarcke random field model must conform to the condition of stationary random field in mathematical sense. Secondly, the soil mass is a collection of infinite points, and the experimental data are the measured values of individual points. If the analysis results of the experimental points in a borehole are used to reflect the properties of the surrounding soil mass, the data used should have ergodic properties of various states. It can be seen that whether the spatial distribution of soil properties is stationary and ergodic is the key to the application of random field method in geotechnical engineering.”
Question 5: It is suggested to add the correlation analysis on the engineering properties of marine soft soil with high water moisture content and its influence on actual engineering, so as to make the research purposemore scientific and practical.
Answer: Thank you for your advice,about the engineering properties of marine soft soil with high water moisture content and its influence on actual engineering, We have made a supplement in the introduction, and the specific content is “In engineering construction, due to the nature of marine soft soil, engineering design and construction can not achieve the desired results, resulting in a large number of diseases and security risks, causing engineering accidents. Further strengthening the research on marine soft soil has important practical significance for reducing and preventing engineering accidents. In the study of soft soil layer, the accurate acquisition of soft soil parameters is a very important prerequisite.” It's on page 1 of the article, line 32-37.
For the minor comments, we have checked this information and made changes where appropriate
We tried our best to improve the manuscript and made some changes in the manuscript. These changes will not influence the content and framework of the paper. And here we did not list the changes but marked in revised paper. We appreciate for Reviewers warm work earnestly, and hope that the correction will meet with approval. Once again, thank you very much for your comments and suggestions.

Reviewer 2 Report
Your article contains a large number of lexical, grammatical, spelling errors, along with ugly design in latex. All this shows the carelessness of the authors to editors, reviewers and readers. Due to this fact, your article is perceived as low quality. Although if we discard all these flaws, on the whole the work is decent, accompanied by a sufficient amount of theoretical information.
All comments and mistakes you can found in attached PDF file.

Author Response
Dear Reviewer, thank you for reviewing our manuscript and for the constructive comments, which greatly helped us to improve the manuscript. The manuscript was carefully revised and point-by-point response was listed below. We hope that your comments have been addressed accurately.
Question 1:Fix English, use "are stationary and ergodic" or rephrase.(page 1 line 14)
Answer: Change “are stationarity and ergodicity” to “are stationary and ergodic”.
Question 2:Is it necessary? (page1 line 28)
Answer: Delete “and is thick,”.
Question 3: Use a synonym.
Answer: Changed to “taked” in page 2 line 42.
Question 4:You have reference 13, please use cite it. (page2 line 43)
Answer: References were quoted and the serial number was updated to [7].
Question 5:Replace one with "investigation". (page2 line 47)
Answer: Replace with “investigation”.
Question 6:Use citation of your references here. (page1 line 57)
Answer: References [14-17].
Question 7:Probably "stationarity". (page2 line 85)
Answer: Changed to “stationarity”
Question 8:Is it t(h)? Because you do not mention r(h) anywhere before. (page3 line 89)
Answer: Changed to “x(h)”.
Question 9:Please check translation, maybe other term should be used. (page3 line 91)
Answer:“weak smoothness” change to “weakly stationary”.
Question 10:Rephrase with different words. (page3 line 91-92)
Answer: “When the geotechnical parameters are detrended term” change to “After detrending treatment of geotechnical parameters”.
Question 11:Should be subscript? (page4 line 119)
Answer: RX is autocorrelation of X should be subscript.
Question 12:What is this? Fix. (page4 line 126, 130)
Answer: Changed “” to “=”.
Question 13:u is subscript. (page4 line 137)
Answer: Changed to “δu”.
Question 14:Choose another formatting. (page 4 line 140-142)
Answer: Changed to “I、II、III、IV”.
Question 15:Better use "find" or "determine". (page4 line 155)
Answer: Changed to “find”.
Question 16:Please use latex table from MDPI article template.
Answer: All table are used latex table from MDPI article template.
Question 17:"Mathematical expression" or just "Formula" (Table 1)
Answer: Used “Mathematical expression”.
Question 18:Capital (Table 1)
Answer: Changed to “Correlation”.
Question 19:", m" or " (m)" please use consistently throughout the text.
Answer: All units changed to “(m)”.
Question 20:Please make same formatting here and in (13). (page 6 line 184)
Answer: Changed to “”, consistent with formula (13).
Question 21:w is subscript. (page 6 line 187)
Answer: Changed to “sw(h)”.
Question 22:Insert space. (page 6 line 194)
Answer: Add a space before the unit “m”.
Question 23:Capital?(page 6 line 196)
Answer: Changed to “Huaiyin - Xiangshuikou”.
Question 24:characteristics. (Table 2)
Answer: Changed to “characteristics”.
Question 25:the. (Table 2)
Answer: Changed to “the”.
Question 26:"angle". (Table 2)
Answer: Changed to “angle”.
Question 27:Recommend you to choose another notation, because comma is not obvious, may be use asterisk - * or other index. (Table 2)
Answer: Used “*”.
Question 28:Use citation of your references here. (page 9 line 251)
Answer: Add and cite references [30].
Question 29:Use citation of your references here. (page 9 line 254)
Answer: Referenced [19, 20].
Question 30:superscript. (page 9 line 260)
Answer: Changed to “h2”.
Question 31:Use the past tense. (page 15 line 471)
Answer: Changed to “studied”.
Question 32:Write contributions. (page 16 line 502-505)
Answer: Added “Author Contributions: Conceptualization, H.L. and X.M.; Funding acquisition, H.L.; Investigation, H.L. and X.M.; Methodology, H.L.; Validation, X.M.; Visualization, X.M. and H.L.; Writing—original draft, H.L. and X.M.; Writing—review & editing, X.M. and H.L. All authors have read and agreed to the published version of the manuscript.”
Question 33:Fill out these. (page 16 line 506-510)
Answer: Added “Funding: This study was partially supported with research grants from the major projects of natural science research in Higher Education Institutions of Anhui Province (KJ2019ZD11), Open Fund of State Key Laboratory of Coal Resources and Safe Mining (No. SKLCRSM20KFA06)and the National Natural Science Foundation of China (41977253). The data used in this paper can be accessed by contacting the corresponding author directly.”
Question 34:Please declare something here. (page 16 line 518)
Answer: Added “Conflicts of Interest: Authors have no conflict of interest to declare.”
We tried our best to improve the manuscript and made some changes in the manuscript. These changes will not influence the content and framework of the paper. And here we did not list the changes but marked in red in revised paper. We appreciate for Reviewers warm work earnestly, and hope that the correction will meet with approval. Once again, thank you very much for your comments and suggestions.

This manuscript is a resubmission of an earlier submission. The following is a list of the peer review reports and author responses from that submission.